# Training Shallow and Thin Networks for Acceleration via Knowledge Distillation with Conditional Adversarial Networks

**Zheng Xu**
University of Maryland
College Park, USA
`xuzh@cs.umd.edu`

**Yen-Chang Hsu**
Georgia Institute of Technology
Atlanta, USA
`yenchang.hsu@gatech.edu`

**Jiawei Huang**
Honda Research Institute
Mountain View, USA
`jhuang@honda-ri.com`

## Abstract

There is an increasing interest on accelerating neural networks for real-time applications. We study the student-teacher strategy, in which a small and fast student network is trained with the auxiliary information learned from a large and accurate teacher network. We propose to use conditional adversarial networks to learn the loss function to transfer knowledge from teacher to student. The experiments on three different image datasets show the student network gain a performance boost with proposed training strategy.

## 1 Introduction

Deep neural networks (DNNs) achieve massive success in artificial intelligence by substantially improving the state-of-the-art performance in various applications. The accuracy of DNNs for large-scale image classification has become comparable to humans on several benchmark datasets (Russakovsky et al., 2015). The recent progress towards such impressive accomplishment is largely driven by exploring deeper and wider network architectures (He et al., 2016; Zagoruyko & Komodakis, 2016). However, it is difficult to deploy the trained modern networks on embedded systems for real-time applications because of the heavy computation and memory cost. In the meantime, the demand for low cost networks is increasing for applications on mobile devices and autonomous cars.

In previous studies, knowledge transfer has been used to train shallow but wide student networks, which potentially have more parameters than the teacher networks (Ba & Caruana, 2014; Urban et al., 2017); ensemble of networks are used as teacher, and a student network with similar architecture and capacity can be trained (Hinton et al., 2015).

In this paper, we focus on improving the performance of a shallow and thin modern network (student) by learning from the dark knowledge (Hinton et al., 2015) of a deep and wide network (teacher). Both the student and teacher networks are convolutional neural networks (CNNs) with residual connections, and the student network is shallow and thin so that it can run much faster than the teacher network during inference. Instead of adopting the classic student-teacher strategy of forcing the output of a student network to exactly mimic the soft targets produced by a teacher network, we introduce conditional adversarial networks to transfer knowledge from teacher to student. We empirically show that the loss learned by the adversarial training has the advantage over the predetermined loss in the student-teacher strategy, especially when the student network has relatively small capacity.

## 2 Learning loss with adversarial networks

Instead of forcing the student to exactly mimic the teacher by minimizing KL-divergence (Hinton et al., 2015), the knowledge is transferred from teacher to student through a discriminator in our approach. This discriminator is trained to distinguish whether the output logits is from teacher or student network, while the student is adversarially trained to fool the discriminator, i.e., output logits that are indistinguishable to the teacher logits.

**Discriminator update.** We denote the discriminator that predicts binary value "Real/Fake" as $D(\cdot)$. To train $D$, we fix the student network $F(\cdot)$ and seek to maximize the log-likelihood, which is known as binary cross-entropy loss,

$$\mathcal{L}_A(D, F) = \frac{1}{N} \sum_{i=1}^{N} \Big( \log P(\text{Real}|D(t_i)) + \log P(\text{Fake}|D(F(x_i))) \Big). \tag{1}$$

The plain adversarial loss $\mathcal{L}_A$ for knowledge distillation, which follows the original GAN (Goodfellow et al., 2014), faces two major challenges. First, the adversarial training process is difficult. Even if we replace the log-likelihood with advanced techniques such as Wasserstein GAN (Arjovsky et al., 2017) or Least Squares GAN (Mao et al., 2016), the training is still slow and unstable in our experiments. Second, the discriminator captures the high-level statistics of teacher and student outputs, but the low-level alignment is missing. The student outputs $F(x_i)$ for $x_i$ can be aligned to a completely unrelated teacher sample $t_j$ by optimizing $\mathcal{L}_A$, which means a dog image can generate a logits vector that predicts cat. One extreme example is that the student always mispredicts dog as cat and cat as dog, but the overall output distribution may still be close to the teacher's.

To tackle these problems, we modify the discriminator objective to also predict the class labels, where the output of discriminator $D(\cdot)$ is a $C + 2$ dimensional vector with C *Label* predictions and a *Real/Fake* prediction. We now maximize

$$\mathcal{L}_{\text{Discriminator}}(D, F) = \frac{1}{2}(\mathcal{L}_A(D, F) + \mathcal{L}_{DS}(D, F)), \tag{2}$$

where $\mathcal{L}_A$ is the previously defined adversarial loss over *Real/Fake*, $\mathcal{L}_{DS}$ is the supervised log-likelihood of discriminator over labels $l_i$, written as

$$\mathcal{L}_{DS}(D, F) = \frac{1}{N} \sum_{i=1}^{N} \Big( \log P(l_i|D(t_i)) + \log P(l_i|D(F(x_i))) \Big). \tag{3}$$

**Student update:** We update the student network after updating the discriminator in each iteration. When updating the student network $F(\cdot)$, we aim to fool the discriminator by fixing discriminator $D(\cdot)$ and minimizing the adversarial loss $\mathcal{L}_A$. In the meantime, the student network is also trained to satisfy the auxiliary classifier of discriminator $\mathcal{L}_{DS}$. Besides the category-level knowledge in $\mathcal{L}_{DS}$, we introduce instance-level knowledge by aligning outputs of teacher and student,

$$\mathcal{L}_{L_1}(F) = \frac{1}{N} \sum_{i=1}^{N} \|F(x_i) - t_i\|_1. \tag{4}$$

The $L_1$ norm has been found helpful in the GAN-based approach for image to image translation (Isola et al., 2017).

Finally, we combine the learned loss with the classification loss $\mathcal{L}_S$ (cross entropy loss) on target categories, and minimize the following objective for the student network $F(\cdot)$,

$$\mathcal{L}_{\text{Student}}(D, F) = \mathcal{L}_S(F) + \mathcal{L}_{L_1}(F) + \mathcal{L}_{GAN}(D, F),$$
$$\text{where } \mathcal{L}_{GAN}(D, F) = \frac{1}{2}(\mathcal{L}_A(D, F) - \mathcal{L}_{DS}(D, F)). \tag{5}$$

The sign of $\mathcal{L}_{DS}$ is flipped in (2) and (5) because both the discriminator and student are trained to preserve the category-level knowledge.

## 3 EXPERIMENTS

We consider three image classification datasets: ImageNet32 (Chrabaszcz et al., 2017), CIFAR-10 and CIFAR-100 (Krizhevsky & Hinton, 2009), and use wide residual networks (WRNs) (Zagoruyko & Komodakis, 2016) for both student and teacher networks. The teacher network is a fixed WRN-40-10, while the student network is shallower and thiner. We use multi-layer preceptron (MLP) with three layers as the discriminator in our approach. To speed up the experiments, the logits of teacher network are generated offline and stored in memory. We use stochastic gradient descent (SGD) as optimizer and follow standard training scheduler, and set dropout ratio to 0.3 for both discriminator and student networks. The results are the median of five random runs.

|  | CIFAR-10 | CIFAR-100 | ImageNet32 |
|---|---|---|---|
| Teacher-net | 4.19 | 20.62 | 38.41 |
| Student-net | 7.46 | 28.52 | 48.2 |
| Student-net + KD (T=1) | 7.27 | 28.62 | 49.37 |
| Student-net + KD (T=2) | 7.3 | 28.33 | 49.48 |
| Student-net + KD (T=5) | 7.02 | 27.06 | 49.63 |
| Student-net + KD (T=10) | 6.94 | 27.07 | 51.12 |
| Student-net + Ours | **6.09** | **25.75** | **47.39** |

Table 1: Error rate achieved on benchmark datasets.

Table 1 shows the error rate of classification on the three benchmark datasets. The teacher is the deep and wide WRN-40-10. The student is much shallower and thinner, WRN-10-4 for CIFARs, and WRN-22-4 for ImageNet32. We choose a larger student network for ImageNet32 because it contains more samples and categories. The first two rows of Table 1 show the performance of standard supervised learning for student and teacher networks, without knowledge transfer. We then compare our approach with knowledge distillation (KD) (Hinton et al., 2015). We choose the temperature parameter $T \in \{1, 2, 5, 10\}$ following the original work. No parameter is tuned for our method. The proposed method improves the performance of small network for all three datasets, and outperforms KD by a margin.

| Loss composition | CIFAR-10 | CIFAR-100 |
|---|---|---|
| $\mathcal{L}_S$ | 7.46 | 28.52 |
| $\mathcal{L}_{GAN}$ | 14.82 | 47.04 |
| $\mathcal{L}_S + \mathcal{L}_{GAN}$ | 6.56 | 27.27 |
| $\mathcal{L}_S + \mathcal{L}_{L_1}$ | 6.44 | 26.66 |
| $\mathcal{L}_S + \mathcal{L}_{L_1} + \mathcal{L}_{GAN}$ | **6.09** | **25.75** |

Table 2: Ablation study for the effect of different components of proposed loss. The numbers are error rate.

Next, we look into the effect of enabling and disabling different components of the proposed approach, as shown in Table 2. By combining the adversarial loss and the category-level knowledge transfer (Equation (2)), the learned loss $\mathcal{L}_{GAN}$ performs reasonably well. However, the indirect knowledge provided by $\mathcal{L}_{GAN}$ alone is not as good as standard supervised learning $\mathcal{L}_S$. Both category-level knowledge transferred by $\mathcal{L}_{GAN}$ and instance-level knowledge transferred by $\mathcal{L}_{L_1}$ can improve the performance of training student network. Our final approach combines these components and performs the best.

## 4  CONCLUSION

We study the student-teacher strategy for network acceleration in this paper. We propose to use adversarial networks to learn the loss for transferring knowledge from teacher to student. We show that the proposed approach can improve the learning of student network, especially when the student network is shallow and thin. Furthermore, in our full analysis which is not included here, we find that it is possible to train a student that is 7x smaller and 5x faster than the teacher without loss of accuracy. Therefore it is a direction worth further exploration.

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
