# OpenReview forum: "Training Shallow and Thin Networks for Acceleration via Knowledge Distillation with Conditional Adversarial Networks"
_ICLR.cc/2018/Workshop — Accept_

### Official Review · AnonReviewer2 · 2018-03-09

**Rating:** 6
**Confidence:** 4

**Review:**

Authors investigate knowledge distillation (KD) in order to train a shallow and thin  student network from a deeper and wider teacher network. Instead of minimizing the KL between the teacher and student, authors propose to match the output logits using adversarial training.  They evaluate their propose approach on ImageNet32, CIFAR10 and CIFAR100 where their approach outperforms KD.

Using adversarial training in a KD setting appears novel to me. However, authors are to use a combination of adversarial loss with L1 and cross-entropy to ease the optimization of models. In their ablation study it is not clear how important is the GAN loss as the use of cross entropy + L1 leads already to good results.

---

### Official Review · AnonReviewer1 · 2018-03-09

**Rating:** 6
**Confidence:** 4

**Review:**

Summary: This paper describes using GAN loss instead of minimizing KL-divergence between teacher & students to make student network better. The authors show it is a helpful on small datasets, and improves distillation result based on Student + L1 method.

Overall it is an interesting paper. My concern is whether this method still works on larger dataset. Many algorithms works well on small network won't work
well on large dataset. So if the author is able to demo result on standard ImageNet level dataset it will be more convincing.

Question: Why add adversarial loss only on logit, instead of deep features?

---

### Comment · AnonReviewer3 · 2018-03-14
**An interesting method to improve the teacher-student model for the purpose of training deep learning model with simpler structure**

This paper presents an interesting method to utilize the auxiliary information from a larger and accurate teacher network.
The  GAN idea has been adapted to train a discriminator model D(.) to help generalize a network with smaller size (i.e., the student network).
The proposed method is verified on 3 datasets and it shows the superior performance when compared with the work from Hinton for the knowledge distillation.
The key for the performance improvements comes from both the L1 norm loss as in (4) and the GAN loss as in (5). The L1 norm enforces that the output from the student network should be close to the output from the teachers, while the loss in (5) learns a discriminator to discriminate between the Fake/Real output from the student.
It seems that the teacher output t_i should be fixed during the whole process, and is it helpful to train the teacher network simultaneously?
I think it maybe more clearer if the authors could use a figure to show the structure of the proposed loss structure.
This learning setting is actually quite similar to the learning using privileged learning (LUPI) setting, and the authors should also discuss some relationships between knowledge distillation and LUPI.

Vladimir Vapnik, Akshay Vashist:
A new learning paradigm: Learning using privileged information. Neural Networks 22(5-6): 544-557 (2009)

Xinxing Xu, Joey Tianyi Zhou, Ivor W. Tsang, Zheng Qin, Rick Siow Mong Goh, Yong Liu:
Simple and Efficient Learning using Privileged Information. CoRR abs/1604.01518 (2016)

---

> ### Author Response · Authors · 2018-03-16
> **thanks for comments**
>
> We thank the reviewer for the insightful comments. We will improve the manuscript based on the suggestions, and we provide response to specific questions below.
>
> Q: It seems that the teacher output t_i should be fixed during the whole process, and is it helpful to train the teacher network simultaneously?
> A: Teacher ouputs are fixed in the proposed method. We are not sure if it is helpful to train the teacher. Wang et al. 2016 observes the benefits of jointly training. However, training together may harm the performance of both teacher and student if they agree on the same bad solution. Moreover, training teacher requires much more computation and memory cost.
>
> Wang, J.; Wei, Z.; Zhang, T.; and Zeng, W. 2016. Deeply-fused nets. arXiv preprint arXiv:1605.07716
>
>
> Q: This learning setting is actually quite similar to the learning using privileged learning (LUPI) setting, and the authors should also discuss some relationships between knowledge distillation and LUPI.
> A: Thanks for the reference. We agree that it is similar to LUPI in a broad sense that they all trained with information that is not used in inference. We will discuss more details in the next version and an extended draft. We feel that in-depth discussion of knowledge distillation and LUPI is slightly beyond the scope of this workshop manuscript. We briefly discuss some relationship here,
> 1) LUPI is a general concept on using extra information for training, while knowledge distillation focus on transfering knowledge from teacher to student.
> 2) Despite LUPI is very general, in a lot of previous works, the auxiliary information provided in training is often extra (e.g. text description of images for image classification task). The auxiliary information in knowledge distillation is embedded in the network structure of teacher and the pre-trained model, which is discovered from the same set of training samples (e.g. images for image classification task).
> 3) Despite LUPI is very general, LUPI is often implemented as variants of SVM+ in previous works, while we focus on knowledge distillation for neural networks.

---

### Decision · Program_Chairs · 2018-03-20
**ICLR 2018 Workshop Acceptance Decision**

**Decision:**

Accept

**Comment:**

Congratulations, your paper was accepted to the ICLR workshop.